# FinRL-Meta: Market Environments and Benchmarks for Data-Driven Financial Reinforcement Learning

**Xiao-Yang Liu**[1*], **Ziyi Xia**[1*], **Jingyang Rui**[2], **Jiechao Gao**[3], **Hongyang Yang**[1],
**Ming Zhu**[4], **Christina Dan Wang**[5†], **Zhaoran Wang**[6], **Jian Guo**[7†]

[1]Columbia University; [2]The University of Hongkong; [3]University of Virginia;
[4]SIAT CAS; [5]New York University (Shanghai); [6]Northwestern University;
[7]IDEA Research, International Digital Economy Academy
{xl2427, zx2325, hy2500}@columbia.edu, christina.wang@nyu.edu,
zhaoranwang@northwestern.edu, guojian@idea.edu.cn

## Abstract

Finance is a particularly challenging playground for deep reinforcement learning. However, establishing high-quality market environments and benchmarks for financial reinforcement learning is challenging due to three major factors, namely, low signal-to-noise ratio of financial data, survivorship bias of historical data, and backtesting overfitting. In this paper, we present an openly accessible FinRL-Meta library that has been actively maintained by the AI4Finance community. First, following a DataOps paradigm, we will provide hundreds of market environments through an automatic data curation pipeline that processes dynamic datasets from real-world markets into gym-style market environments. Second, we reproduce popular papers as stepping stones for users to design new trading strategies. We also deploy the library on cloud platforms so that users can visualize their own results and assess the relative performance via community-wise competitions. Third, FinRL-Meta provides tens of Jupyter/Python demos organized into a curriculum and a documentation website to serve the rapidly growing community. FinRL-Meta is available at: https://github.com/AI4Finance-Foundation/FinRL-Meta

## 1 Introduction

Finance is a particularly challenging playground for deep reinforcement learning (DRL) [1, 2], including investigating market fragility [3], designing profitable strategies [4, 5, 6], and assessing portfolio risk [7, 8]. However, establishing near-real market environments and benchmarks on financial reinforcement learning are difficult due to three major factors, namely, low signal-to-noise ratio (SNR) of financial data, survivorship bias of historical data, and backtesting overfitting. Such a *simulation-to-reality gap* [9, 10] degrades the performance of DRL strategies in real markets. Therefore, high-quality market environments and benchmarks are crucial for the research and industrialization of data-driven financial reinforcement learning.

Existing works have applied various DRL algorithms in financial applications [7, 11, 12, 13]. Many of them have reported better trading performance in terms of cumulative return and Sharpe ratio. Several recent works [7, 14, 12, 15] showed the great potential of DRL-based market simulators. However, these works are difficult to reproduce. The FinRL library [16, 11] provided an open-source framework for financial reinforcement learning. However, it focused on the reproducibility of backtesting performance by providing several market environments. A workshop version of

---

[*]Equal contribution.
[†]Corresponding authors.

36th Conference on Neural Information Processing Systems (NeurIPS 2022).

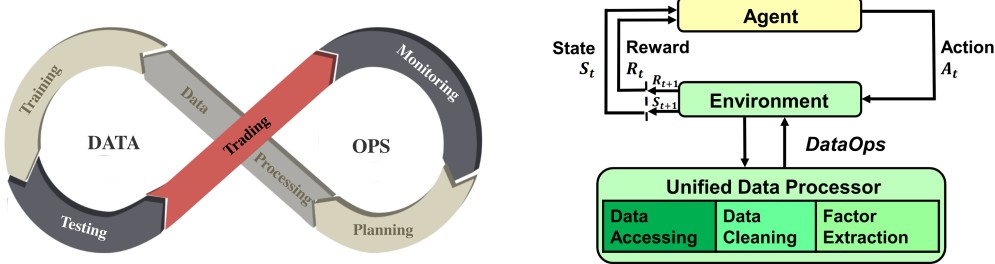

Figure 1: DataOps paradigm (left) and data-driven financial reinforcement learning (right).

FinRL-Meta [17] provided data processors to access and clean unstructured market data, but it did not provide benchmarks back then.

The DataOps paradigm [18, 19, 20] refers to a set of practices, processes, and technologies that combines automated data engineering and agile development [20]. It helps reduce the cycle time of data engineering and improve data quality. To deal with financial big data (usually unstructured), we follow the DataOps paradigm and implement an automatic pipeline in Fig. 1(left): task planning, data processing, training-testing-trading, and monitoring agents' performance. Through this pipeline, we are able to continuously produce DRL benchmarks on dynamic market datasets.

In this paper, we present an openly accessible FinRL-Meta library that has been actively maintained by the AI4Finance community. We aim to create an infrastructure to enable real-time paper trading and facilitate the real-world adoption of financial reinforcement learning. This is relevant to the broader RL research community since it provides a rare case of a task that can be tested against real-world performance without major investment, while robotics requires simulation or expensive equipment and games are available in simulations.

Fig. 1(right) shows an overview of data-driven financial reinforcement learning. First, following a DataOps paradigm [18, 19, 20], we provide hundreds of market environments through an automatic data curation pipeline that collects dynamic datasets from real-world markets and processes them into standard gym-style market environments. Second, we reproduce popular papers as benchmarks, including high-frequency stock trading, cryptocurrency trading and stock portfolio allocation, serving as stepping stones for users to design new strategies. With the help of the data curation pipeline, we hold our benchmarks on cloud platforms so that users can visualize their own results and assess the relative performance via community-wise competitions. Third, FinRL-Meta provides dozens of Jupyter/Python demos as educational materials, organized in a curriculum, and a documentation website to serve the rapidly growing community.

The remainder of this paper is organized as follows. Section 2 reviews existing works. Section 3 describes challenges and presents an overview of our FinRL-Meta framework. Section 4 describes how we process data into market environments. In Section 5, we benchmark popular DRL papers. Finally, we conclude this paper in Section 6.

## 2 Related Works

We review DataOps practices and existing works on data-driven reinforcement learning.

**DataOps practices**: DataOps [20, 19, 18] applies the ideas of lean development and DevOps to the data science field. DataOps practices have been developed in companies and organizations to improve the quality and efficiency of data analytics [19]. These implementations consolidate various data sources, unify and automate the pipeline of data analytics, including data accessing, cleaning, analysis, and visualization.

However, the DataOps methodology has not been applied to financial reinforcement learning researches. Most researchers access data, clean data, and extract technical indicators (features) in a case-by-case manner, which involves heavy manual work and may not guarantee the data quality.

**Data-driven reinforcement learning**: Environments are crucial for training DRL agents [1].

Table 1: List of state space, action space, and reward function.

| Key components | Attributes |
|---|---|
| State | Balance $b_t \in \mathbb{R}_+$; Shares $\boldsymbol{h}_t \in \mathbb{Z}_+^n$ |
| | Opening/high/low/close price $\boldsymbol{o}_t, \boldsymbol{h}_t, \boldsymbol{l}_t, \boldsymbol{p}_t \in \mathbb{R}_+^n$ |
| | Trading volume $\boldsymbol{v}_t \in \mathbb{R}_+^n$ |
| | Fundamental indicators; Technical indicators |
| | Social data; Sentiment data |
| | Alpha and beta signals; Smart beta indexes, etc. |
| Action | Buy/Sell/Hold |
| | Short/Long |
| | Portfolio weights |
| Reward | Change of portfolio value |
| | Portfolio log-return |
| | Sharpe ratio |
| Environments | Dow-30, S&P-500, NASDAQ-100 |
| | Cryptocurrencies |
| | Foreign currency and exchange |
| | Futures; Options; ETFs; Forex |
| | CN securities; US securities |
| | Paper trading; Live Trading |

- **OpenAI gym** [21] provides standardized environments for a collection of benchmark problems that expose a common interface, which is widely supported by many libraries [22, 23, 24]. Three trading environments, TradingEnv, ForexEnv, and StocksEnv, are included to support stock and FOREX markets. However, it has not been updated for years.

- **D4RL** [25] introduces the idea of *datasets for deep data-driven reinforcement learning* (D4RL). It provides benchmarks in offline RL. However, D4RL does not provide financial environments.

- **FinRL** [16, 11] is an open-source library that builds a full pipeline for financial reinforcement learning. It contains three market environments, i.e., stock trading, portfolio allocation, crypto trading, and two data sources, i.e., Yahoo Finance and WRDS. However, those market environments of FinRL cannot meet the community's growing demands.

- **NeoRL** [26] collected offline RL environments for four areas, CityLearn [27], FinRL [16, 11], Industrial Benchmark [28], and MuJoCo [29], where each area contains several gym-style environments. Regarding financial aspects, it directly imports market environments from FinRL.

**Benchmarks of financial reinforcement learning**: Many researches applied DRL algorithms in quantitative finance [4, 5, 6, 30, 14, 31] by building their own market environments. Despite the above-mentioned open-source libraries that provide some useful environments, there are no established benchmarks yet. On the other hand, the data accessing, cleaning and factor extraction processes are usually limited to data sources like Yahoo Finance and Wharton Research Data Services (WRDS).

## 3 Financial Reinforcement Learning and FinRL-Meta Framework

We describe financial reinforcement learning and its challenges, then provide an overview of our FinRL-Meta framework.

### 3.1 Financial Reinforcement Learning and Challenges

Assuming full observability, we model a trading task as a Markov Decision Process (MDP) with five tuples [1] $(\mathcal{S}, \mathcal{A}, \mathbb{P}, r, \gamma)$, where $\mathcal{S}$ and $\mathcal{A}$ denote the state space and action space, respectively, $\mathbb{P}(s'|s, a)$ is the transition probability of an unknown environment, $r(s, a, s')$ is a reward function, and $\gamma \in (0, 1]$ is a discount factor. A trading agent learns a policy $\pi(s_t|a_t)$ that maximizes the discounted cumulative return $R = \sum_{t=0}^{T} \gamma^t r(s_t, a_t, s_{t+1})$ over a trading period $t = 0, 1, ..., T$. Detailed modelling can be found in [4]

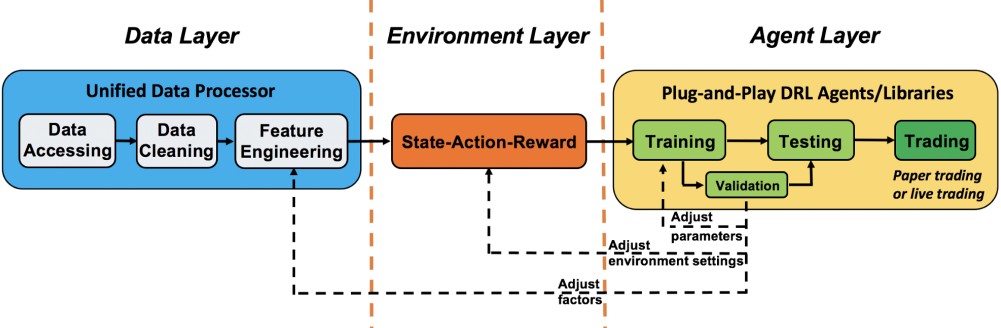

Figure 2: Overview of FinRL-Meta framework.

The historical dataset before time $t = 0$ is used to train the trading agent. Note that we process the dataset into a market environment, following the *de facto* standard of OpenAI gym [21]. In Table 1, we list the state space, action space, and reward function.

- **State** $s \in \mathcal{S}$: A state represents an agent's perception of a market situation, which may include balance, shares, OHLCV values, technical indicators, social data, sentiment data, etc.

- **Action** $a \in \mathcal{A}$: An action is taken from the allowed action set at a state. Actions may vary for different trading tasks, e.g., for stock trading, the actions are the number of shares to buy/sell for each stock, while for portfolio allocation, the actions are the allocation weights of the capital.

- **Reward** $r(s, a, s')$: Reward is an incentive mechanism for an agent to learn a better policy. Several common reward functions are provided: 1). Change of portfolio value $r(s, a, s') = v' - v$, where $v'$ and $v$ are portfolio values at state $s'$ and $s$, respectively; 2). Portfolio log return $r(s, a, s') = \log(v'/v)$; and 3). Sharpe ratio [32] defined in Section 5.1.

The above full observability assumption can be extended to partial observation (the underlying states cannot be directly observed), i.e., partially observable Markov Decision Process (POMDP). A POMDP model utilizes a Hidden Markov Model (HMM) [33] to model a time series that is caused by a sequence of unobservable states. Considering the noisy financial data, it is natural to assume that a trading agent cannot directly observe market states. Studies suggested that the POMDP model can be solved by using recurrent neural networks, e.g., an off-policy Recurrent Deterministic Policy Gradient (RDPG) algorithm [34], and a long short-term memory (LSTM) network that encodes partial observations into a state of a reinforcement learning algorithm [35].

Training and testing environments based on historical data may not simulate real markets accurately due to the *simulation-to-reality gap* [9, 10], and thus a trained agent cannot be directly deployed in real-world markets. We summarize three major factors for the *simulation-to-reality gap* in financial reinforcement learning as follows:

- **Low signal-to-noise ratio (SNR) of financial data**: Data from different sources may contain large noise [36] such as random noise, outliers, etc. It is challenging to identify alpha signals or build smart beta indexes using noisy datasets.

- **Survivorship bias of historical market data**: Survivorship bias is caused by a tendency to focusing on existing stocks and funds without consideration of those that are delisted [37]. It could lead to an overestimation of stocks and funds, which will mislead the agent.

- **Backtesting overfitting**: Existing research mainly report backtesting results. It is highly possible that authors are tempted to tune hyper-parameters and retrain the agent multiple times [3] to obtain better backtesting results, resulting in model overfitting [38, 39].

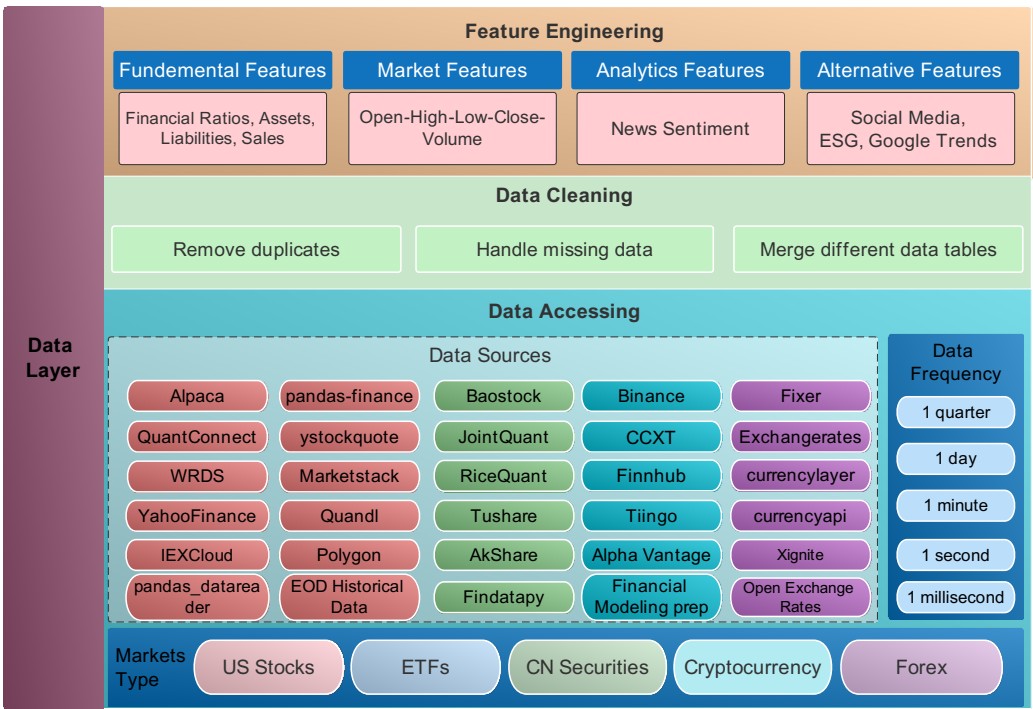

Figure 3: Data layer of FinRL-Meta.

## 3.2 Overview of FinRL-Meta

FinRL-Meta builds a universe of market environments for data-driven financial reinforcement learning. FinRL-Meta follows the *de facto* standard of OpenAI Gym [21] and the *lean principle* of software development. It has the following unique features.

**Layer structure and extensibility**: As shown in Fig. 2, we adopt a layered structure that consists of three layers, data layer, environment layer, and agent layer. Layers interact through end-to-end interfaces, achieving high extensibility. For updates and substitutes inside a layer, this structure minimizes the impact on the whole system. Moreover, the layer structure allows easy extension of user-defined functions and fast updating of algorithms with high performance.

**Training-testing-trading pipeline**: We employ a training-testing-trading pipeline that the DRL approach follows a standard end-to-end pipeline. The DRL agent is first trained in a training environment and then fined-tuned (adjusting hyperparameters) in a validation environment. Then the validated agent is tested on historical datasets (backtesting). Finally, the tested agent will be deployed in paper trading or live trading markets. Refer to Appx.E for more details.

**Plug-and-play mode**: In the above training-testing-trading pipeline, a DRL agent can be directly plugged in, then trained and tested. The following DRL libraries are supported:

- **ElegantRL [24]**: Lightweight, efficient and stable algorithms using PyTorch.
- **Stable-Baselines3 [22]**: Improved DRL algorithms based on OpenAI Baselines.
- **RLlib [23]:** An open-source DRL library that offers high scalability and unified APIs.

## 4 Financial Big Data and Data Curation Pipeline for Dynamic Datasets

Financial big data is usually unstructured in format. We process four types of data [39] into market environments, including fundamental data (e.g., earning reports), market data (e.g., OHLCV data), analytics (e.g., news sentiment), and alternative data (e.g., social media data, ESG data).

---

[3]There is information leakage.

### 4.1 Data Layer for Unstructured Financial Big Data

**Data curation for data-driven financial reinforcement learning**: We follow the DataOps paradigm [20, 19, 18] in the data layer. As shown in Fig. 3, we establish a standard pipeline for financial data engineering, which processes data from different sources into a unified market environment following the *de facto* standard of OpenAI gym [21]. We automate this pipeline with a data processor that implements the following functionalities:

- **Data accessing**: Users can connect data APIs of different market platforms via our common interfaces. Users can access data agilely by specifying the start date, end date, stock list, time interval, and other parameters. FinRL-Meta has supported more than 30 data sources, covering stocks, cryptocurrencies, ETFs, forex, etc.

- **Data cleaning**: Raw data retrieved from different data sources are usually of various formats and with erroneous or missing data to different extents. It makes data cleaning highly time-consuming. With a data processor, we automate the data cleaning process. In addition, we use stock ticker name and data frequency as unique identifiers to merge all types of data into a unified data table.

- **Feature engineering**: In feature engineering, FinRL-Meta aggregates effective features which can help improve model predictive performance. We provide various types of features, including but not limited to fundamental, market, analytics, and alternative features. Users can quickly add features using open-source libraries or add user-defined features. Users can add new features in two ways: 1) Write a user-defined feature extraction function directly. The returned features are added to a feature array. 2) Store the features in a file, and put it in a default folder. Then, an agent can read these features from the file.

**Automated feature engineering**: FinRL-Meta currently supports four types of features:

- **Fundamental features**: Fundamental features are processed based on the earnings data in SEC filings queried from WRDS. The data frequency is low, typical quarterly, e.g., four data points in a year. To avoid information leakage, we use a two-month lag beyond the standard quarter end date, e.g., Apple released its earnings report on 2022/07/28 for the third quarter (2022/06/25) of year 2022. Thus for the quarter between 04/01 and 06/30, our trade date is adjusted to 09/01 (same method for other three quarters). We also provide functions in our data processor for calculating financial ratios based on earnings data such as earnings per share (EPS), return on asset (ROA), price to earnings (P/E) ratio, net profit margin, quick ratio, etc.

- **Market features**: Open-high-low-close price and volume data are the typical market data we can directly get from querying the data API. They have various data frequencies, such as daily prices from YahooFinance, TAQ (Millisecond Trade and Quote) from WRDS. In addition, we automate the calculation of technical indicators based on OHLCV data by connecting the Stockstats[4] or TA-lib library[5] in our data processor, such as Moving Average Convergence Divergence (MACD), Average Directional Index (ADX), Commodity Channel Index (CCI), etc.

- **Analytics features**: We provide news sentiment for analytics features. First, we get the news headline and content from WRDS [40]. Next, we use NLTK.Vader[6] to calculate sentiment based on the sentiment compound score of a span of text by normalizing the emotion intensity (positive, negative, neutral) of each word. For the time alignment with market data, we use the exact enter time, i.e., when the news enters the database and becomes available, to match the trade time. For example, if the trade time is every ten minutes, we collect the previous ten minutes' news based on the enter time; if no news is detected, then we fill the sentiment with 0.

- **Alternative features**: Alternative features are useful, but hard-to-obtain from different data sources [39], such as ESG data, social media data, Google trend searches, etc. ESG (Environmental, social, governance) data are widely used to measure the sustainability and societal impacts of an investment. The ESG data we provide is from the Microsoft Academic Graph database, which is an open resource database with records of scholar publications. We have functions in our data processor to extract AI publication and patent data, such as paper citations, publication counts, patent counts, etc. We believe these features reflect companies' research and development capacity for AI technologies [41, 42]. It is a good reflection of ESG research commitment.

---

[4]Github repo: https://github.com/jealous/stockstats
[5]Github repo: https://github.com/mrjbq7/ta-lib
[6]Github repo: https://github.com/nltk/nltk

## 4.2 Environment Layer for Creating Dynamic Market Environments

FinRL-Meta follows the OpenAI gym-style [21] to create market environments using the cleaned data from the data layer. It provides hundreds of environments with a common interface. Users can build their environments using FinRL-Meta's interfaces, share their results and compare a strategy's trading performance. Following the gym-style [21], each environment has three functions as follows:

- `reset()` function resets the environment back to the initial state $s_0$

- `step()` function takes an action $a_t$ from the agent and updates state from $s_t$ to $s_{t+1}$.

- `reward()` function computes the reward value transforming from $s_t$ to $s_{t+1}$ by action $a_t$.

Detailed descriptions can be found in [5][38].

We plan to add more environments for users' convenience. For example, we are actively building market simulators using limit-order-book data (refer to Appx.C.7), where we simulate the market from the playback of historical limit-order-book-level data and an order matching mechanism. We foresee the flexibility and potential of using a Hidden Markov Model (HMM) [33] or a generative adversarial net (GAN) [43] to generate market scenarios [31].

**Incorporating trading constraints to model market frictions**: To better simulate real-world markets, we incorporate common market frictions (e.g., transaction costs and investor risk aversion) and portfolio restrictions (e.g., non-negative balance).

- **Flexible account settings**: Users can choose whether to allow buying on margin or short-selling.

- **Transaction cost**: We incorporate the transaction cost to reflect market friction, e.g., $0.1\%$ of each buy or sell trade.

- **Risk-control for market crash**: In FinRL [16, 11], a turbulence index [44] is used to control risk during market crash situations. However, calculating the turbulence index is time-consuming. It may take minutes, which is not suitable for paper trading and live trading. We replace the financial turbulence index with the volatility index (VIX) [45] that can be accessed immediately.

**Multiprocessing training via vector environment**: We utilize GPUs for multiprocessing training, namely, the vector environment technique of Isaac Gym [46], which significantly accelerates the training process. In each CUDA core, a trading agent interacts with a market environment to produce transitions in the form of {state, action, reward, next state}. Then, all the transitions are stored in a replay buffer and later used to update a learner. By adopting this technique in our market simulator, we successfully achieve the multiprocessing simulation of hundreds of market environments to improve the performance of DRL trading agents on large datasets.

## 4.3 Advantages

Our data curation pipeline is automatic, which gives us the following three advantages.

**Curriculum for newcomers**: We provide an educational curriculum, as shown in Fig. 4, for community newcomers with different levels of proficiency and learning goals. Users can grow programming skills by gradually changing the data/environment layer following instructions on our website.

**Benchmarks on cloud**: We provide demos on a cloud platform, Weights & Biases [7], to demonstrate the training process. We define the hyperparameter sweep, training function, and initialize an agent to train and tune hyperparameters. On the cloud platform Weights & Biases, users are able to visualize their results and assess the relative performance via community-wise competitions.

**Curriculum learning for agents**: Based on FinRL-Meta (a universe of market environments, say $\geq 100$), one is able to construct an environment by sampling data samples from multiple market datasets, similar to XLand [47]. In this way, one can apply the curriculum learning method [47] to train a generally capable agent for several financial tasks.

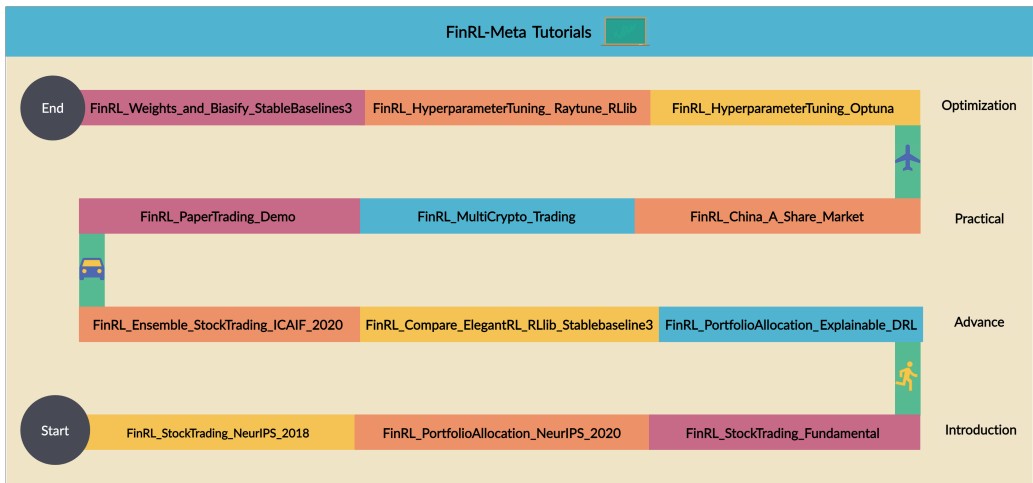

Figure 4: Demos of FinRL-Meta, organized in a curriculum structure.

# 5    Tutorials and Benchmarks of Financial Reinforcement Learning

We provide dozens of tutorial notebooks to serve as stepping stones for newcomers and reproduce popular papers as benchmarks for follow-up research.

## 5.1    Metrics and Baselines for Evaluating Performance

We provide the following metrics to measure the trading performance:

- **Cumulative return** $R = \frac{v - v_0}{v_0}$, where $v$ is the final portfolio value, and $v_0$ is the original capital.
- **Annualized return** $r = (1 + R)^{\frac{365}{t}} - 1$, where $t$ is the number of trading days.
- **Annualized volatility** $\sigma_a = \sqrt{\frac{\sum_{i=1}^{n}(r_i - \bar{r})^2}{n-1}}$, where $r_i$ is the annualized return in year $i$, $\bar{r}$ is the average annualized return, and $n$ is the number of years.
- **Sharpe ratio** [32] $S_T = \frac{\text{mean}(R_t) - r_f}{\text{std}(R_t)}$, where $R_t = \frac{v_t - v_{t-1}}{v_{t-1}}$, $r_f$ is the risk-free rate, and $t = 1, ..., T$.
- **Max. drawdown**: The maximal percentage loss in portfolio value.

The following baseline trading strategies are provided for comparisons:

- **Passive trading strategy** [48] is a well-known long-term strategy. The investors just buy and hold selected stocks or indexes without further activities.
- **Mean-variance and min-variance strategy** [49] are two widely used strategies that look for a balance between risks and profits. They select a diversified portfolio in order to achieve higher profits at a lower risk.
- **Equally weighted strategy** is a portfolio allocation strategy that gives equal weights to different assets, avoiding allocating overly high weights on particular stocks.

## 5.2    Tutorials and Demos in Jupyter Notebooks

For educational purposes, we provide Jupyter notebooks as tutorials[8] to help newcomers get familiar with the whole pipeline.

- **Stock trading** [4]: We apply popular DRL algorithms to trade multiple stocks.

---

[7]Website: https://wandb.ai/site
[8]https://github.com/AI4Finance-Foundation/FinRL-Tutorials

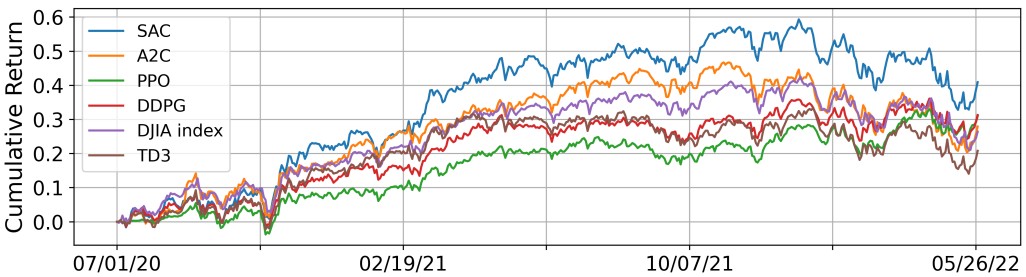

Figure 5: Reproducing stock trading of [4].

- **Portfolio allocation** [16]: We use DRL agents to optimize asset allocation in a set of stocks.
- **Cryptocurrency trading** [16]: We reproduce the experiment [16] on 10 popular cryptocurrencies.
- **Multi-agent RL for liquidation strategy analysis** [8]: We reproduce the experiment in [8]. The multi-agent optimizes the shortfalls in the liquidation task, which is to sell given shares of one stock sequentially within a given period, considering the costs arising from the market impact and the risk aversion.
- **Ensemble strategy for stock trading** [5]: We reproduce the experiment in [5] that employed an ensemble strategy of several DRL algorithms on the stock trading task.
- **Paper trading demo**: We provide a demo for paper trading. Users could combine their own strategies or trained agents in paper trading.
- **China A-share demo**: We provide a demo based on the China A-share market data.
- **Hyperparameter tuning**: We provide several demos for hyperparameter tuning using Optuna [50] or Ray Tune [51], since hyperparameter tuning is critical for better performance.

### 5.3 Reproducing Prior Papers as Benchmarks

We have reproduced experiments in several papers as benchmarks. Users can study our codes for research purpose or use them as stepping stones for deploying trading strategies in live markets. In this subsection, we describe three home-grown examples specifically. For more benchmarks, please refer to Appx. C.

**Stock trading task** [4]: We access Yahoo! Finance database and select the 30 constituent stocks (accessed at 07/01/2020) in Dow Jones Industrial Average (DJIA). We use data from 01/01/2009 to 06/30/2020 for training and data from 07/01/2020 to 05/31/2022 for testing. We use technical indicators in our state space, e.g., Moving Average Convergence Divergence (MACD), Relative Strength Index (RSI), Commodity Channel Index (CCI), Average Directional Index (ADX), etc.

As shown in Fig. 5, we train five popular DRL algorithms to trade and compare their results with the DJIA index. We show a detailed walkthrough of how DRL works in the stock trading task, on which many subsequent works are based [4]. This benchmark is beneficial for getting into the field of financial reinforcement learning.

**Podracer on the cloud** [52, 53]: We reproduce cloud solutions of population-based training, e.g., generational evolution [52] and tournament-based evolution [53]. If GPUs are abundant, users can take advantage of this benchmark to work on high-frequency trading tasks. Detailed instructions are provided on our website.

**Ensemble strategy** [5]: The ensemble method combines different agents to obtain an adaptive one, which inherits the best features of agents and performs remarkably well in practice. We consider three component algorithms, Proximal Policy Optimization (PPO), Advantage Actor-Critic (A2C), and Deep Deterministic Policy Gradient (DDPG), which have different strengths and weaknesses. For instance, A2C is good at dealing with a bearish trend market. PPO is good at following trends and acts well in generating more returns in a bullish market. DDPG can be used as a complementary strategy to PPO in a bullish trend. Using a rolling window, an ensemble agent automatically selects

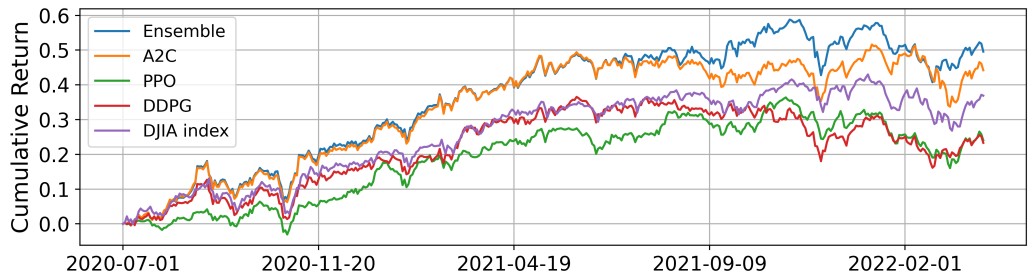

Figure 6: Reproducing the ensemble strategy of [5]: cumulative return.

| (2020/07/01-2022/03/31) | Ensemble [5] | A2C | PPO | DDPG | DJIA index |
|---|---|---|---|---|---|
| Annual Return | 25.9% | 23.3% | 13.1% | 12.7% | 19.7% |
| Annual Volatility | 15.9% | 16.2% | 13.4% | 15.0% | 14.4% |
| Sharpe Ratio | 1.53 | 1.37 | 0.99 | 0.88 | 1.32 |
| Calmar Ratio | 2.27 | 1.97 | 0.88 | 0.85 | 1.74 |
| Max Drawdown | -11.4% | -11.8% | -14.9% | -14.9% | -11.3% |

Table 2: Reproducing the ensemble strategy of [5].

the best model for each test period. Again on the 30 constituent stocks of the DJIA index, we use data from 04/01/2009 to 06/30/2019 for training, and data from 07/01/2020 to 03/31/2022 for validation and testing through a quarterly rolling window.

From Fig. 6 and Table 2, we observe that the ensemble agent outperforms other agents. In the experiment, the ensemble agent has the highest Sharpe ratio of 1.53, which means it performs the best in balancing risks and profits. This benchmark demonstrates that the ensemble strategy is effective in constructing a more reliable agent based on several component DRL agents.

## 6 Conclusion

In this paper, we obeyed the DataOps paradigm and developed a FinRL-Meta library that provides openly accessible dynamic financial datasets and reproducible benchmarks. For future work, FinRL-Meta aims to build a universe of financial market environments, like the XLand environment [47]. To improve the performance for large-scale markets, we are exploiting GPU-based massive parallel simulation such as Isaac Gym [46] and deployed it into projects such as RL for market simulator. Moreover, it will be interesting to explore the evolutionary perspectives [54, 55, 52, 53] to simulate the markets. We believe that FinRL-Meta will provide insights into complex market phenomena and offer guidance for financial regulations.

## Acknowledgement

We thank Mr. Tao Liu (IDEA Research, International Digital Economy Academy) for technical support of computing platform on this research project. Ming Zhu was supported by National Natural Science Foundations of China (Grant No. 61902387). Christina Dan Wang is supported in part by National Natural Science Foundation of China (NNSFC) grant 11901395 and Shanghai Pujiang Program, China 19PJ1408200.

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
