Table 3: List of key terms for reinforcement learning.

| Key Terms | Description |
|---|---|
| Agent [56] | A decision maker |
| Environment [56] | A world with which an agent interacts with |
| Gym-style environment [21] | A standard form of DRL environment by OpenAI |
| Markov Decision Process (MDP) | A mathematical framework to model decision-making problems |
| State, Action, Reward | Three main factors in an agent-environment interaction |
| Policy | A rule that agent follow to make decision |
| Policy gradient | An approach to solve RL problems by optimizing the policy directly |
| Deep Q-Learning (DQN) [57] | The first DRL algorithm that uses a neural network to approximate the Q-function |
| DDPG [58] | Deep Deterministic Policy Gradient algorithm |
| PPO [59] | Proximal Policy Optimization algorithm |
| Hyperparameter tuning | Change hyperparameter during training to get a converged result faster |
| Ensemble strategy | An ML technique. Here we combine several DRL agent to a better model |
| Population-based training (PBT) | Optimise a population of models and hyperparameters, and select the optimal set |
| Generational evolution [52] | Employing an evolution strategy over generations |
| Tournament-based evolution [53] | An evolution by asynchronously updating a tournament board of models |
| Curriculum learning | Train an ML model from easier to harder data, imitating the human curriculum |
| Simulation-to-reality gap | The difference between simulation environment and real-world task |

Table 4: List of key terms for finance.

| Key Terms | Description |
|---|---|
| Algorithmic trading | A method of trading using designed algorithm instead of human traders |
| Backtesting | A method to see how a strategy performs on a certain period of historical data |
| Signal-to-noise ratio (SNR) | A ratio of desired signal (good data) to undesired signal (noise) |
| DataOps | A series of principles and practices to improve the quality of data science |
| Sentiment data | A category in financial big data that contains subjective viewpoints |
| Historical data | All kinds of data that already existed in the past |
| Survivorship bias | A bias caused by only seeing existed examples, but not those already died out |
| Information leakage | When the data contains future information, causing model overfitting |
| Paper trading | Simulation of buying and selling without using real money |
| OHLCV | A popular form of market data with: Open, High, Low, Close, Volume |
| Technical indicators | A statistical calculation based on OHLCV data to indicate future price trends |
| Market frictions | A financial market friction as anything that interferes with trade. |
| Market crash | A huge drop of market price within a very short time |
| Volatility index (VIX) [45] | A market index that shows the market's expectations for volatility |
| Limit Order Book (LOB) | A list to record the interest of buyers and sellers |
| Smart beta index | An enhanced indexing strategy to beat a benchmark index |
| Liquidation, trade execution | An investor closes their position in an asset |

# A   Terminology of Reinforcement Learning and Finance

We provide a list of key terms and corresponding descriptions for reinforcement learning and finance in Table 3 and Table 4.

For terminologies of reinforcement learning, interested users can refer to [56] or the classic textbook [1]. Also, the webpage[9] explains key concepts of RL.

For terminologies of finance, interested users can refer to [39].

---

[9]OpenI SpinningUp: `https://spinningup.openai.com/en/latest/spinningup/rl_intro.html`

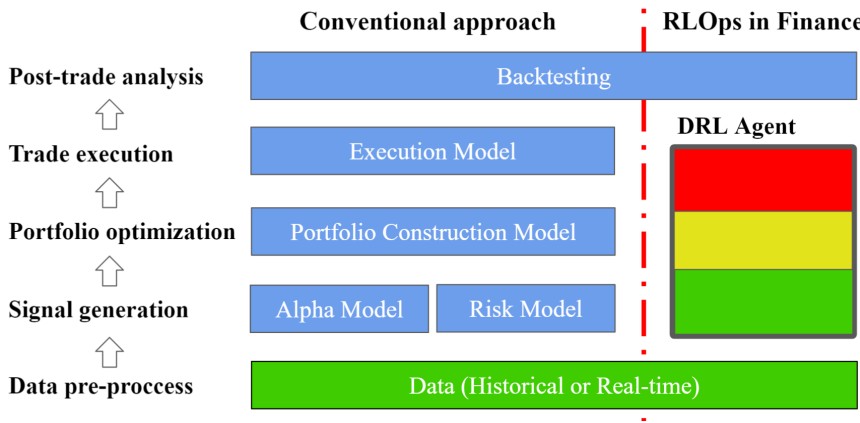

Figure 7: Comparison between conventional machine learning approach and RLOps in finance for an algorithmic trading process. (This figure is from [52].)

# B  DataOps Paradigm for Financial Big Data

The DataOps paradigm [20], or more accurately the methodology, is a way of organizing people, processes and technology to deliver reliable and high quality data quickly to all its users. It helps reduce the cycle time of data engineering and improves data quality. The practice of DataOps focuses on enabling collaboration across the organization to drive agility, speed of delivery and new data initiatives. By leveraging the power of automation, DataOps aims to address the challenges associated with inefficiencies in access, preparation, integration and availability of data.

However, the DataOps approach has not been applied to research in financial reinforcement learning. Most researchers acquire data, clean it and extract technical indicators (features) on a case-by-case manner, which involves heavy manual work and may not guarantee high data quality.

To handle the unstructured financial big data, FinRL-Meta follows the DataOps paradigm and implement a data curation pipeline in Fig. 1 (left): task planning, data processing, training-testing-trading, and monitoring the performance of the agents.

- The first step is task planning, such as stock trading, portfolio allocation, cryptocurrency trading, etc.
- Then, we do data processing, including data accessing and cleaning, and feature engineering.
- Next step is where RL takes part in. In particular, the training-testing-trading process in Fig. 10.
- The final step is performance monitoring.

Through this pipeline, FinRL-Meta replaces this process with a single step and can continuously process dynamic market datasets into market environments.

# C  FinRL: Financial Reinforcement Learning

In this appendix, we take a practical perspective to provide an overview of financial reinforcement learning. First, we explain the paradigm of *RLOps in fiance* [52] that may help deploy and maintain RL trading agents in real-world markets reliably and efficiently. Then, we selectively describe several applications along with practical demos. For applications of RL in finance, interested readers are suggested to read [2] that provides a complete survey of various applications, including optimal execution, portfolio optimization, option pricing and hedging, market making, smart order routing, and robo-advising.

## C.1  RLOps Paradigm in Finance

Algorithmic trading [60, 61] has been widely adopted in financial investments. The lifecycle of a conventional machine learning strategy may include five general stages, as shown in Fig. 7 (left),

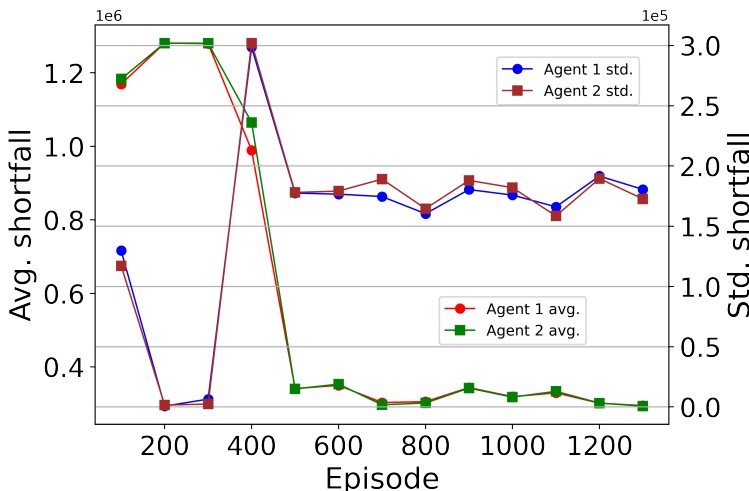

Figure 8: Liquidation analysis of [8].

namely data pre-processing, modeling and trading signal generation, portfolio optimization, trade execution, and post-trade analysis. Recently, deep reinforcement learning (DRL) [62, 63, 1] has been recognized as a powerful approach for quantitative finance, since it has the potential to overcome some important limitations of supervised learning, such as the difficulty in label specification and the gap between modeling, positioning, and order execution.

We advocate extending the principle of *MLOps* [64][10] to the *RLOps in finance* paradigm that implements and automates the continuous training (CT), continuous integration (CI), and continuous delivery (CD) for trading strategies. We argue that such a paradigm has vast profits potential from a broadened horizon and fast speed, which is critical for wider DRL adoption in real-world financial tasks. The *RLOps in finance* paradigm, as shown in Fig. 7 (right), integrates middle stages (i.e., modeling and trading signal generation, portfolio optimization, and trade execution) into a DRL agent. Such a paradigm aims to help quantitative traders develop an end-to-end trading strategy with a high degree of automation, which removes the latency between stages and results in a compact software stack. The major benefit is that it can explore the vast potential profits behind the large-scale financial data, exceeding the capacity of human traders; thus, the trading horizon is lifted into a potentially new dimension. Also, it allows traders to continuously update trading strategies, which equips traders with an edge in a highly volatile market. However, the large-scale financial data and fast iteration of trading strategies bring imperative challenges in terms of computing power.

### C.2  Stock Trading

Referring to [4] and [65], we access Yahoo! Finance database and select 30 constituent stocks (accessed at 07/01/2020) in the Dow Jones Industrial Average (DJIA) index. We use data from 01/01/2009 to 06/30/2020 for training and data from 07/01/2020 to 05/31/2022 for testing. The following technical indicators are used in the state space: Moving Average Convergence Divergence (MACD), Relative Strength Index (RSI), Commodity Channel Index (CCI), Average Directional Index (ADX), etc.

Code available at: https://github.com/AI4Finance-Foundation/FinRL/blob/master/tutorials/1-Introduction/FinRL_StockTrading_Fundamental.ipynb

### C.3  Liquidation Analysis and Trade Execution

Reproducing [8], We build a simulated environment of stock prices according to the Almgren and Chriss model. Then we implement the multi-agent DRL algorithms for both competing and cooperative liquidation strategies. This benchmark demonstrates the trade execution task using deep

---

[10]MLOps is an ML engineering culture and practice that aims at unifying ML system development (Dev) and ML system operation (Ops).

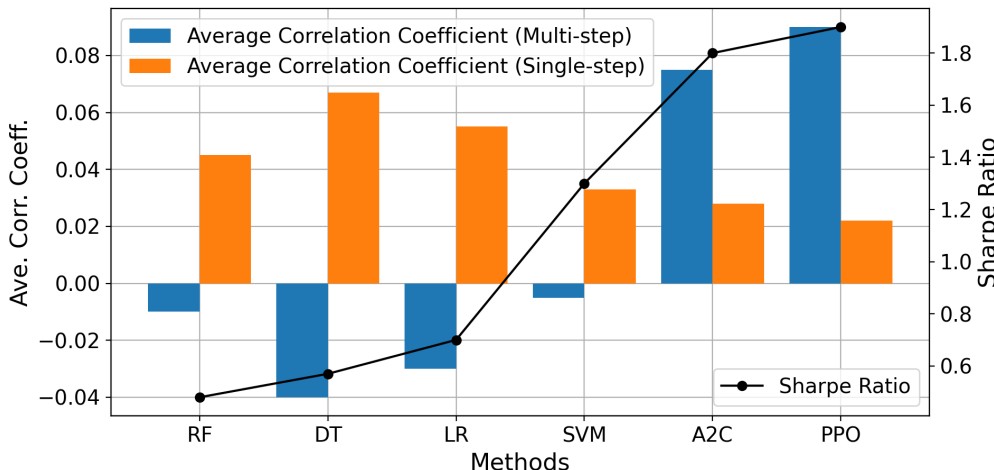

Figure 9: Reproducing portfolio management of [66]: Comparison of average correlation coefficient and Sharpe ratio among ML and DRL methods.

reinforcement learning algorithms. When trading, traders want to minimize the expected trading cost, which is also called implementation shortfall. In Fig. 8, there are two agents, and we see that the implementation shortfalls decrease during the training process.

Code available at: https://github.com/AI4Finance-Foundation/FinRL-Meta/tree/master/tutorials/2-Advance/execution_optimizing

### C.4 Explainable Financial Reinforcement Learning

We reproduce [66] that compares the performance of DRL algorithms with machine learning (ML) methods on the multi-step prediction in the portfolio allocation task. We use four technical indicators MACD, RSI, CCI, and ADX as features. Random Forest (RF), Decision Tree Regression (DT), Linear Regression (LR), and Support Vector Machine (SVM) are the ML algorithms in comparison. We use data from Dow Jones 30 constituent stocks to construct the environment. We use data from 04/01/2009 to 03/31/2020 as the training set and data from 04/01/2020 to 05/31/2022 for backtesting. In Fig. 9, the results show that DRL methods have higher Sharpe ratio than ML methods. Also, DRL methods' average correlation coefficient are significantly higher than that of ML methods (multi-step).

We reproduce [66] that compares the performance of DRL algorithms with machine learning (ML) methods on the multi-step prediction in the portfolio allocation task. We use four technical indicators MACD, RSI, CCI, and ADX as features. Random Forest (RF), Decision Tree Regression (DT), Linear Regression (LR), and Support Vector Machine (SVM) are the ML algorithms in comparison. We use data from Dow Jones 30 constituent stocks to construct the environment. We use data from 04/01/2009 to 03/31/2020 as the training set and data from 04/01/2020 to 05/31/2022 for backtesting.

Code available at: https://github.com/AI4Finance-Foundation/FinRL/blob/master/tutorials/2-Advance/FinRL_PortfolioAllocation_Explainable_DRL.ipynb

### C.5 Podracer on the Cloud

We reproduce cloud solutions of population-based training, e.g., generational evolution [52] and tournament-based evolution [53]. FinRL-Podracer can easily scale out to $\geq$ 1000 GPUs, which features high scalability, elasticity and accessibility by following the cloud-native principle.

If GPUs are abundant, users can take advantage of this benchmark to meet the real-time requirement of high-frequency trading tasks. Detailed instructions are provided on our website.

On an NVIDIA SuperPOD cloud, we conducted extensive experiments on stock trading, and found that it substantially outperforms competitors, such as OpenAI and RLlib [52].

Code available at: `https://github.com/AI4Finance-Foundation/FinRL_Podracer`

### C.6   Ensemble Strategy

The ensemble method combines different agents to obtain an adaptive one, which performs remarkably well in practice. We consider three component algorithms, Proximal Policy Optimization (PPO), Advantage Actor-Critic (A2C), and Deep Deterministic Policy Gradient (DDPG), which have different strengths and weaknesses. Using a rolling window, an ensemble agent automatically selects the best model for each test period. Again on the 30 constituent stocks of the DJIA index, we use data from 04/01/2009 to 06/30/2019 for training, and data from 07/01/2020 to 03/31/2022 for validation and testing through a quarterly rolling window.

Code available at: `https://github.com/AI4Finance-Foundation/FinRL/blob/master/tutorials/2-Advance/FinRL_Ensemble_StockTrading_ICAIF_2020.ipynb`

### C.7   Market Simulator [15]

**Gym market environments [21]**

We build all of our environments following OpenAI-gym style. A first reason is this makes convenience for plugging in any of the three DRL libraries (Stable Baseline3, RLlib, ElegantRL). Another reason is user friendly. Newcomers can learn our environments faster and build their own task-specific environments.

**Synthetic data generation**:

We simulate the market from the playback of historical limit-order-book-level data and the order matching mechanism. Currently, we simulate at the minute level (i.e. one time step = one minute), which is changeable. The state is a stack of market indicators and market snapshots from the last few time steps. The action is to place an order. We support market orders and limit orders. We also provide several wrappers to accept typically discrete or continuous actions. Rewards can be configured by the participants with the aim of generating policies that optimize pre-specified indicators. In our simulator, we take into account the following factors: 1) temporary market impact; 2) order delay. We do not consider the following factors in our simulator: 1) permanent market impact of limit orders; 2) non-resiliency limit order book.

Code available at: `https://github.com/AI4Finance-Foundation/Market_Simulator`

# D  Dataset Documentation and Usages

We organize the dataset documentation according to the suggested template of *datasheets for datasets* [11].

## D.1  Motivation

- **For what purpose was the dataset created?**

  As data is refreshing minute-to-millisecond, finance is a particularly difficult playground for deep reinforcement learning.

  In academia, scholars use financial big data to obtain more complex and precise understanding of markets and economics. While industries use financial big data to refine their analytical strategies and strengthen their prediction models. To serve the rapidly growing FinRL community, we creates FinRL-Meta that provides data accessing from different sources and build the data to RL environments. We aim to provide dynamical RL environments that are manageable by users.

  We aim to build a financial metaverse, a universe of near real-market environments, as a playground for data-driven financial machine learning.

- **Who created the dataset?**

  FinRL-Meta is an open-source project created by the FinRL community. Contents of FinRL-Meta are contributed by the authors of this paper and will be maintained by members of FinRL community.

- **Who funded the creation of the dataset?**

  AI4Finance Foundation, a non-profit open-source community that shares AI tools for finance, funded our project.

## D.2  Composition

- **What do the instances that comprise the dataset represent?**

  Instances of FinRL-Meta are financial data includes: stocks, securities, cryptocurrencies, etc. FinRL-Meta provides hundreds of market environments through an automatic pipeline that collects dynamic datasets from real-world markets and processes them into standard gym style market environments. FinRL-Meta also benchmarks popular papers as stepping stones for users to design new trading strategies.

- **How many instances are there in total?**

  FinRL-Meta does not store data directly. Instead, we provide codes for data accessing, data cleaning, feature engineering, and building into RL environments. Table D.1 provides the supported data sources of FinRL-Meta.

  At the moment, there are hundreds of market environments, tens of tutorials and demos, and several benchmarks provided.

- **Does the dataset contain all possible instances or is it a sample of instances from a larger set?**

  With our provided codes, users could fetch data from the data source by properly specifying the starting date, ending date, time granularity, asset set, attributes, etc.

- **What data does each instance consist of?**

  Now there are several types of financial data, as shown in Table D.1:

  - Stocks
  - Cryptocurrencies
  - Securities
  - ETFs
  - Exchange rate:

---

[11] Timnit Gebru, Jamie Morgenstern, Briana Vecchione, Jennifer Wortman Vaughan, Hanna Wal- lach, Hal Daumé Iii, and Kate Crawford. Datasheets for datasets. *Communications of the ACM*, 64(12):86–92, 2021.

| Data Source | Type | Max Frequency | Raw Data | Preprocessed Data |
|---|---|---|---|---|
| Alpaca | US Stocks, ETFs | 1 min | OHLCV | Prices, indicators |
| Baostock | CN Securities | 5 min | OHLCV | Prices, indicators |
| Binance | Cryptocurrency | 1 s | OHLCV | Prices, indicators |
| CCXT | Cryptocurrency | 1 min | OHLCV | Prices, indicators |
| IEXCloud | NMS US securities | 1 day | OHLCV | Prices, indicators |
| JoinQuant | CN Securities | 1 min | OHLCV | Prices, indicators |
| QuantConnect | US Securities | 1 s | OHLCV | Prices, indicators |
| RiceQuant | CN Securities | 1 ms | OHLCV | Prices, indicators |
| Tushare | CN Securities | 1 min | OHLCV | Prices, indicators |
| WRDS | US Securities | 1 ms | Intraday Trades | Prices, indicators |
| YahooFinance | US Securities | 1 min | OHLCV | Prices, indicators |
| AkShare | CN Securities | 1 day | OHLCV | Prices, indicators |
| findatapy | CN Securities | 1 day | OHLCV | Prices, indicators |
| pandas_datareader | US Securities | 1 day | OHLCV | Prices, indicators |
| pandas-finance | US Securities | 1 day | OHLCV | Prices, indicators |
| ystockquote | US Securities | 1 day | OHLCV | Prices, indicators |
| Marketstack | 50+ countries | 1 day | OHLCV | Prices, indicators |
| finnhub | US Stocks, currencies, crypto | 1 day | OHLCV | Prices, indicators |
| Financial Modeling prep | US stocks, currencies, crypto | 1 min | OHLCV | Prices, indicators |
| EOD Historical Data | US stocks, and ETFs | 1 day | OHLCV | Prices, indicators |
| Alpha Vantage | Stock, ETF, forex, crypto, technical indicators | 1 min | OHLCV | Prices, indicators |
| Tiingo | Stocks, crypto | 1 day | OHLCV | Prices, indicators |
| Quandl | 250+ sources | 1 day | OHLCV | Prices, indicators |
| Polygon | US Securities | 1 day | OHLCV | Prices, indicators |
| fixer | Exchange rate | 1 day | Exchange rate | Exchange rate, indicators |
| Exchangerates | Exchange rate | 1 day | Exchange rate | Exchange rate, indicators |
| Fixer | Exchange rate | 1 day | Exchange rate | Exchange rate, indicators |
| currencylayer | Exchange rate | 1 day | Exchange rate | Exchange rate, indicators |
| currencyapi | Exchange rate | 1 day | Exchange rate | Exchange rate, indicators |
| Open Exchange Rates | Exchange rate | 1 day | Exchange rate | Exchange rate, indicators |
| XE | Exchange rate | 1 day | Exchange rate | Exchange rate, indicators |
| Xignite | Exchange rate | 1 day | Exchange rate | Exchange rate, indicators |

Table 5: Supported data sources. OHLCV means open, high, low, and close prices; volume data.

- **Is there a label or target associated with each instance?**

  No. There is not label or preset target for each instance. But users can use our benchmarks are baselines.

- **Is any information missing from individual instances?**

  Yes. In several data sources, there are missing values and we provided standard preprocessing methods.

- **Are relationships between individual instances made explicit?**

  Yes. An instance is a sample set of the market of interest.

- **Are there recommended data splits?**

  We recommend users to follow our training-testing-training pipeline, as shown in Fig. 10. Users can flexibly choose their preferred settings, e.g., in stock trading task, our demo access Yahoo! Finance database and use data from 01/01/2009 to 06/30/2020 for training and data from 07/01/2020 to 05/31/2022 for backtesting.

- **Are there any errors, sources of noise, or redundancies in the dataset?**

  For the raw data fetched from different sources, there are noise and outliers. We provide codes to process the data and built them into standard RL gym environment.

- **Is the dataset self-contained, or does it link to or otherwise rely on external resources?**

  It is linked to external resources. As shown in Table D.1, FinRL-Meta fetch data from data sources to build gym environments.

- **Does the dataset contain data that might be considered confidential?**

  No. All our data are from publicly available data sources.

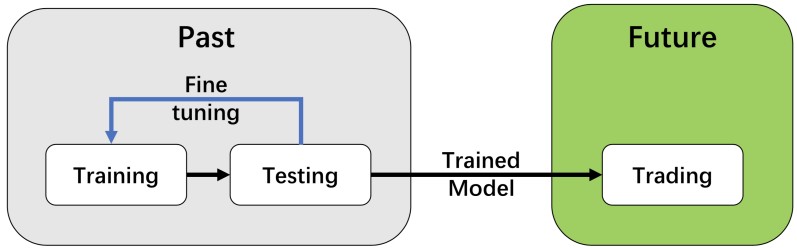

Figure 10: Overview of the training-testing-trading pipeline in FinRL-Meta.

- **Does the dataset contain data that, if viewed directly, might be offensive, insulting, threatening, or might otherwise cause anxiety?**

  No. All our data are numerical.

### D.3 Collection Process

- **How was the data associated with each instance acquired?**

  FinRL-Meta fetch data from data sources. as shown in Table D.1.

- **What mechanisms or procedures were used to collect the data?**

  FinRL-Meta provides dynamic market environments that are built according to users' settings. To achieve this, we provide software APIs to fetch data from different data sources. Note that some data sources requires accounts and passwords or have limitations on number or frequency of requests.

- **If the dataset is a sample from a larger set, what was the sampling strategy?**

  It is dynamic, depending on users' settings, such as the starting date, ending date, time granularity, asset set, attributes, etc.

- **Who was involved in the data collection process and how were they compensated?**

  Our codes collect publicly available market data, which is free.

- **Over what timeframe was the data collected?**

  It is not applicable because the environments are created dynamically by running the codes to fetch data in real time.

- **Were any ethical review processes conducted?**

  No?

### D.4 Preprocessing/cleaning/labeling

- **Was any preprocessing/cleaning/labeling of the data done?**

  Yes. For the raw data fetched from different sources, there are noise and outliers. We provide codes to process the data and built them into standard RL gym environment.

- **Was the "raw" data saved in addition to the preprocessed/cleaned/labeled data**

  The raw data are hold by different data sources (data providers).

- **Is the software that was used to preprocess/clean/label the data available?**

  Yes. We use our own codes to do cleaning and preprocessing.

### D.5 Uses

- **Has the dataset been used for any tasks already?**

  Yes. Thousands of FinRL community members use FinRL-Meta for learning and research purpose. Demos and tutorials are mentioned in Section 5.

- **Is there a repository that links to any or all papers or systems that use the dataset?**

  1. Research papers that used FinRL-Meta are list here:

  ```
  https://github.com/AI4Finance-Foundation/FinRL/blob/master/
  tutorials/FinRL_papers.md
  ```

Our workshop version of FinRL-Meta [17] appeared in NeurIPS 2021 Workshop on Data-Centric AI.

2. The following three repositories has incorporated FinRL-Meta:

- FinRL-Meta corresponding to the market layer of FinRL (5.6K stars):
  https://github.com/AI4Finance-Foundation/FinRL
- ElegantRL (2.1K stars) supports FinRL-Meta:
  https://github.com/AI4Finance-Foundation/ElegantRL
- FinRL-Podracer:
  https://github.com/AI4Finance-Foundation/FinRL_Podracer

- **What (other) tasks could the dataset be used for?**

  Besides the current tasks (tutorial, demo and benchmarks), FinRL-Meta will be useful for the following tasks:

  - **Curriculum learning for agents**: Based on FinRL-Meta (a universe of market environments, say $\geq 100$), one is able to construct a environment by sampling data samples from multiple market datasets, similar to XLand [47]. In this way, one can apply the curriculum learning method [47] to train a generally capable agent for several financial tasks.
  - To improve the performance for the large-scale markets, we are exploiting GPU-based massive parallel simulation such as Isaac Gym [46].
  - It will be interesting to explore the evolutionary perspectives [54, 55, 52, 53] to simulate the markets. We believe that FinRL-Meta will provide insights into complex market phenomena and offer guidance for financial regulations.

- **Is there anything about the composition of the dataset or the way it was collected and preprocessed/cleaned/labeled that might impact future uses?**

  We believe that FinRL-Meta will not encounter usage limit. Our data are fetched from different sources in real time when running the codes. However, there may be one or two out of $\geq 30$ data sources (in Table D.1) change data access rules that may impact future use. So please refer to the rules and accessibility of certain data source when using.

- **Are there tasks for which the dataset should not be used?**

  No. Since there are no ethical problems for FinRL-Meta, users could use FinRL-Meta in any task as long as it does not violate laws.

  **Disclaimer: Nothing herein is financial advice, and NOT a recommendation to trade real money. Please use common sense and always first consult a professional before trading or investing.**

## D.6 Distribution

- **Will the dataset be distributed to third parties outside of the entity (e.g., company, institution, organization) on behalf of which the dataset was created?**

  No. It will always be held on GitHub under MIT license, for educational and research purpose.

- **How will the dataset will be distributed?**

  Our codes and existing environments are available on GitHub FinRL-Meta repository https://github.com/AI4Finance-Foundation/FinRL-Meta.

- **When will the dataset be distributed?**

  FinRL-Meta is publicly available since Feburary 14th, 2021.

- **Will the dataset be distributed under a copyright or other intellectual property (IP) license, and/or under applicable terms of use (ToU)?**

  FinRL-Meta is distributed under MIT License, for educational and research purpose.

- **Have any third parties imposed IP-based or other restrictions on the data associated with the instances?**

  No.

- **Do any export controls or other regulatory restrictions apply to the dataset or to individual instances?**

  No. Our data are fetched from different sources in real time. However, there may be one or two out of $\geq 20$ data sources (in Table D.1) change data access rules that may impact future use. So please refer to the rules and accessibility of certain data source when using.

## D.7 Maintenance

- **Who will be supporting/hosting/maintaining the dataset?**

  FinRL-Meta has been actively maintained by FinRL community (including the authors of this paper) that has over 10K members at the moment. We are still actively updating market environments, to serve the rapidly growing FinRL community.

- **How can the owner/curator/manager of the dataset be contacted?**

  To contact the main developers, we encourage users join our Slack channel:
  https://join.slack.com/t/ai4financeworkspace/shared_invite/
  zt-v67oll1jm-dzTgIT9fHZIjjrqprrY0kg
  or our mailing list:
  https://groups.google.com/u/1/g/ai4finance,

- **Is there an erratum?**

  Users can use GitHub to report issues/bugs, and use Slack channel or mailing list to discuss solutions. FinRL community is actively improving the codes, say extracting technical indicators, evaluating feature importance, quantifying the probability of backtesting overfitting, etc.

- **Will the dataset be updated?**

  Yes, we are actively updating FinRL-Meta's codes and data sources. Users could get information and the newly updated version through our GitHub repository, or join the mailing list: https://groups.google.com/u/1/g/ai4finance.

- **If the dataset relates to people, are there applicable limits on the retention of the data associated with the instances**

  The data of FinRL-Meta do not relate to people.

- **Will older versions of the dataset continue to be supported/hosted/maintained?**

  Yes. All versions can be found on our GitHub repository.

- **If others want to extend/augment/build on/contribute to the dataset, is there a mechanism for them to do so?**

  We maintain FinRL-Meta on GitHub. Users can use GitHub to report issues/bugs, and use Slack channel or mailing list to discuss solutions. We welcome community members to submit pull requests through GitHub.

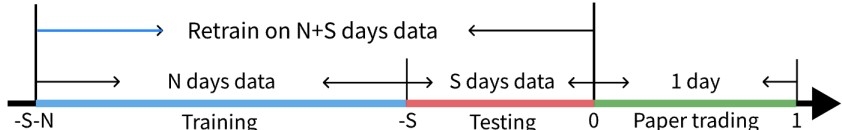

Figure 11: Data split for a window with training, testing and trading.

---

**Algorithm 1** Algorithm for stock paper trading

---

1: Initialize a set of hyper-parameters;
2: **for** $d = 0$ to $D - 1$ **do**
3:     # **Step 1)**. Train an agent
4:     Using data period $[d - S - N, d - S - 1]$ to train an agent,
5:     Using data period $[d - S, d - 1]$ to test the trained agent and adjust hyper-parameters,
6:     # **Step 2)**. Retrain the agent
7:     Using data period $[d - S - N, d - 1]$ to retrain the agent with the adjusted hyper-parameters,
8:     # **Step 3)**. Perform paper trading for one day
9:     Using the trained agent to trade in the $d$-th day.
10: **end for**

---

# E    Dynamic Dataset

For frequently updated financial data, it is crucial to keep the model learning the latest information form the market. However, it will be very time consuming to process data and train the model very frequently. Thus, FinRL-Meta brings forward the concept of dynamic dataset.

The dynamic dataset is a standardized workflow of downloading and processing data following the need of "training-testing-trading" pipeline periodically. This make the time of training controllable.

## E.1    Training-testing-trading pipeline

We employ a training-testing-trading pipeline that the DRL approach follows a standard end-to-end pipeline. The DRL agent is first trained in a training environment and then fined-tuned (adjusting hyperparameters) in a validation environment. Then the validated agent is tested on historical datasets (backtesting). Finally, the tested agent will be deployed in paper trading or live trading markets.

Our construction of dynamic dataset fits the training-testing-trading pipeline well, providing a potential standard work flow of financial reinforcement learning.

## E.2    Paper trading demo

Practical tasks like stock trading and cryptocurrency trading suffer from the false positive issue due to overfitting, where an agent might perform well on testing data but not on real-world markets. Thus, we propose to deploy DRL agent on paper trading.

Fig. 11 shows our "training-testing-trading" pipeline. In a window, there are $N$ days' data for training and $S$ days' data for testing. At the end of a window, we perform paper trading for 1 day. Note that we always retrain the agent using $N + S$ days of training and testing data together. Then, we roll the window forward by 1 day ahead and perform the above steps for a new window. A paper trading is always carried out for 1 day. Therefore, $D$ windows correspond to $D$ trading days.

Alg. 1 summarizes the pipeline of paper trading. For $D$ trading days from 0 to $D - 1$, we keep doing the following three steps:

- **Step 1)**. Download and process $N$-day data, from day $d - S - N$ to day $d - S - 1$. Then build the data into a gym-style environment and train the agent. Then download and process S-day data, from day $d - S$ to day $d - 1$. Then build the data into a gym-style environment and validate how the agent performs. According to the agent's performance on the validation environment, adjust hyper-parameters.

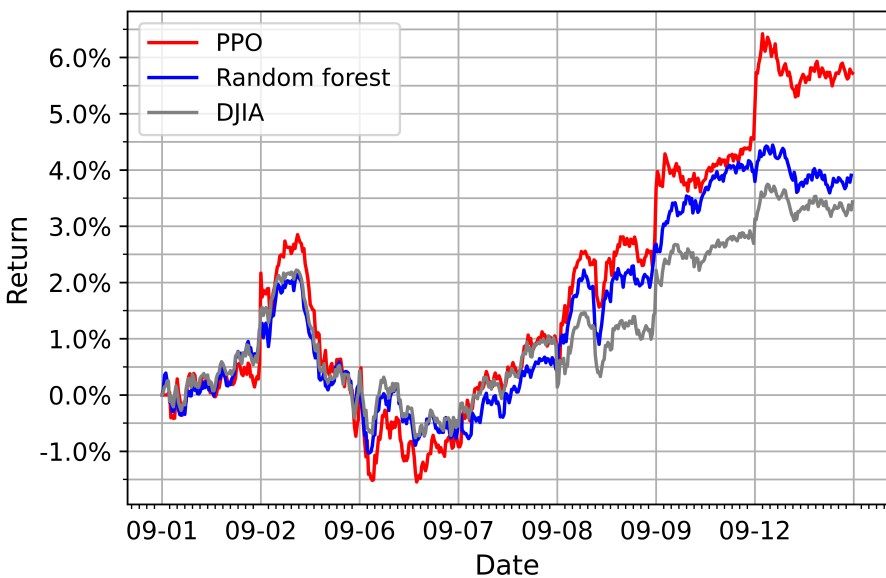

Figure 12: Cumulative returns: a conventional ML method (random forest) VS. our trained PPO agent.

- **Step 2**). Build the training and testing data, totally $N + S$ days from day $d - S - N$ to day $d - 1$ (note that there are 390 data points for each day's minute-level data), into a gym-style environment. Update hyper-parameters to the values chosen from **Step 1**). Then retrain the agent on these $N + S$-day environment.
- **Step 3**). Deploy the trained agent to the paper trading market and trade from 9:30am to 4:00pm.

For the experiment, we select Dow Jones 30 stocks as our trading stocks and use minute-level historical market data from 08/24/2022 to 09/09/2022. The data are downloaded from Alpaca[12]. Then, we use the paper trading APIs provided by Alpaca to do paper trading from 09/01/2022 to 09/12/2022. The cumulative return of our PPO method and conventional random forest method is shown in Fig. 12.

---

[12]Web page of Alpaca: https://alpaca.markets/

# F Open Source FinRL-Meta and DAO, DeFi, NFT, Web3

Over the past decades, we are witnessing capital growth exceeding economic growth globally. However, the door to personal capital growth is not open to all. In a way, one needs to start rich to get richer. The situation is even worsened by the competition with computers. Today in major stock markets, at least $60\%$ of the trades are automated by algorithms.

How to *democratize opportunity* for personal capital growth? We need to ally with the computers to take advantage of unprecedented amounts of data and unparalleled computing infrastructure. Therefore, we come up with this vision to establish an AI framework where retail traders can grow personal capital via a pay-by-use mode. Today, we have reached a full-stack solution that provides financially optimized deep reinforcement learning algorithms together with cloud native solutions. Users can easily access numerous financial data, as well as computational resources whenever needed.

FinRL [16, 11] is the first open-source framework to demonstrate the great potential of financial reinforcement learning [2]. Over several years' development[13], it has evolved into an ecosystem, FinRL-Meta, serving as a playground for data-driven financial reinforcement learning. We believe FinRL-Meta will reshape our financial lives, while our open-source community will make sure it is for the better.

Next, we discuss the potential of combining our open-source FinRL-Meta ecosystem with the emerging technologies, such as DAO, DeFi, NFT[14], and Web3[15].

- Decentralized Autonomous Organizations (DAO) has three features: 1). Member-owned communities without centralized leadership; 2). A safe way to collaborate with Internet strangers; and 3). A safe place to commit funds to a specific cause. In FinRL-Meta's open-source community (over 12K users at the moment of finalizing this paper version), we plan to encourage community members to organize into DAO funds (essentially a distributed fund) in an ad hoc manner. Members of a DAO fund will employ smart contracts to crowdfunding, design strategy, implement algorithmic trading, share profit and loss, etc.

- Decentralized Finance (DeFi) has three features: 1). A global, open alternative to the current financial system; 2). Products that let you borrow, save, invest, trade, and more; and 3). Based on open-source technology that anyone can program with. We would like to encourage community members to actively explore the potential of financial reinforcement learning technologies in the emerging DeFi based financial markets. For example, using the automated environment layer in Section 4.2, we will connect DeFi trading systems into gym-style market environments; also, we are actively working with financial data providers and cloud computing providers to upgrade the trading infrastructure.

- Non-Fungible Token (NFT) has three features: 1). A way to represent anything unique as an Ethereum-based asset; 2). NFTs are giving more power to content creators than ever before; and 3). Powered by smart contracts on the Ethereum blockchain. In FinRL-Meta community, we would like to encourage community members to release trading strategies and codes in the NFT forms, while a DAO fund will buy these NFTs via smart contracts and then trade in a DeFi system.

- Third-Generation of WWW (Web3) incorporates concepts such as decentralization, blockchain technologies, and token-based economics, which is essentially community-run Internet. Our open-source FinRL-Meta community will serve Web3 users a unique infrastructure and playground, where Web3 users can establish DAO funds to trade in a DeFi system.

As a near-term development, FinRL-Meta community would embrace the Massive Open Online Courses (MOOC) paradigm. For demonstrative and education purposes, our community members are actively creating notebooks, blogs and videos. We believe it would be an effective incentive to release them in the form of NFTs. Web3MOOC guarantees authorship and ownership (a right to profit) of those NFTs, thus provides a strong incentive to our community members for content creation.

---

[13]We began to build an open-source community with a practical demonstration [67] at NeurIPS 2018 conference.

[14]Ethereum: https://ethereum.org/en/

[15]Web3 Foundation: https://web3.foundation/

## G  Data Privacy, Strategy Privacy and Federated Learning Technology

As our open-source community is continuously developing new features to ensure that FinRL-Meta provides better user experience, one of the main targets for our next step is to enhance financial data privacy as well as strategy privacy for users. We discuss the potential of integrating the federated learning technology into our open-source FinRL-Meta, in order to achieve data privacy and strategy privacy for our users, say collaboratively training.

Federated learning is a method for training machine learning models from distributed datasets that remains private to data owners. It allows a central machine learning model to overcome the *isolated data island*, i.e., to learn from data sets distributed on multiple devices that do not reveal or share the data to a central server. We understand that in certain scenarios, our users face the problem that they only have a small amount of financial data and are not unable to train a robust model, but they are also hesitate to train their models with others due to privacy reason. To the best of our knowledge, we would like to name a few representative examples on the use of federated learning to help the financial applications. Byrd et al. [68] present a privacy-preserving federated learning protocol on a real-world credit card fraud dataset for the development of federated learning systems. The researchers in WeBank [69] created FATE, an industrial-grade project that supports enterprises and institutions to build machine learning models collaboratively at large-scale in a distributed manner. FATE has been adopted in real-world applications in finance, health and recommender systems. We also want to mention one research work [70] which points out the open problems in federated learning. We believe that, as a newly introduced method, federated learning has a lot of undiscovered and exciting applications that we can develop. We would like to encourage our community members to explore the potential of federated learning technologies for the benefit of financial applications.

# H  Accessibility, Accountability, Maintenance and Rights

FinRL-Meta is an open-source project on GitHub. We use the MIT License for research and educational usage. while users can utilize them as stepping stones for customized trading strategies. Codes, market environments, benchmarks and documentations are available on the GitHub repository: `https://github.com/AI4Finance-Foundation/FinRL-Meta`.

FinRL-Meta has been actively maintained by FinRL community that has over 10K members at the moment. On GitHub, we keep updating our codes, merging pull requests, and fix bugs and issues. We welcome contributions from community members, researchers and quant traders.

We have accumulated six competitive advantages over the past five years. The first three are technology innovations:

- FinRL [16, 11] is the first framework to provide an automatic pipeline for financial reinforcement learning.
- For financial big data, the FinRL-Meta project connects with > 30 market data sources.
- For cloud solutions, the FinRL-Podracer project [52] [53] scales out to $\geq 1000$ GPUs. We have extensive testings on NVIDIA's DGX-2 SuperPod platform.

Based on the above projects and active contributions, an open-source community in the intersection of ML and Finance fields is emerging. Our AI4Finance community is robust with the following three features:

- We have over $10K$ active community members, many of which are actively designing strategies and connecting with paper trading, even live trading. We are collaborating with tens of universities and research institutes, and $\geq 50$ software engineers from big IT companies.
- Both Columbia University (Department of Electrical Engineering, Department of Statistics) and New York University ((Department of Finance) have opened delicate courses about FinRL, while $\geq 120$ students in total have taken it.
- In academia, we have several accepted papers and also delivered several invited talks. Our AI4Finance Foundation (`https://github.com/AI4Finance-Foundation`) serves as a bridge between machine learning, data science, operation research, and finance communities.

As the authors of this paper and core developers of FinRL-Meta, we bear all responsibility in case of violation of rights.

**Disclaimer: Nothing in this paper and the FinRL-Meta repository is financial advice, and NOT a recommendation to trade real money. Please use common sense and always first consult a professional before trading or investing.**

# Financial Reinforcement Learning

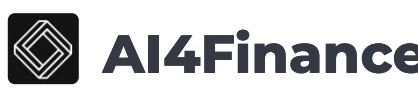 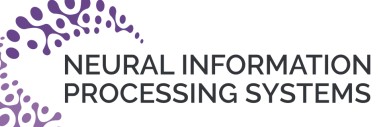

# FinRL

FinRL is the first open-source framework for financial reinforcement learning. We are devoted to developing and sharing ML codes with data scientists, software engineers, quantitative traders, and ML researchers. AI4Finance Foundation aims to democratize the opportunity for personal capital growth and build an **"open finance"** culture.

## Energetic FinRL Ecosystem

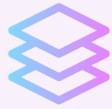

**FinRL**

A full-stack pipeline for algorithmic trading

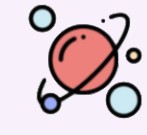

**FinRL-Meta**

A universe of market environments and benchmarks

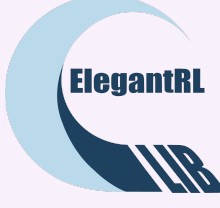

A cloud-native deep reinforcement learning library

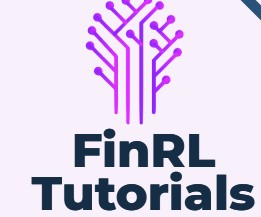

**FinRL Tutorials**

An educational curriculum for all-level users

## Collaborators

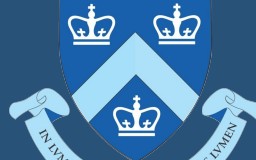
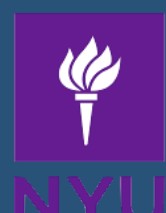

" **DEMOCRATIZE OPPORTUNITY VIA OPEN FINANCE** "

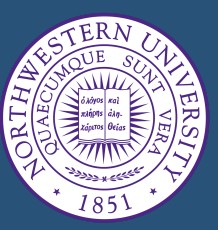
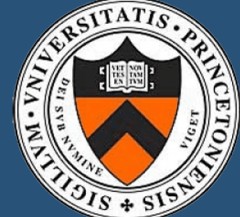

## Open-Finance Community

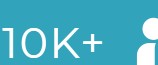 10K+ 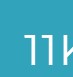 11K 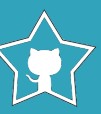 60K 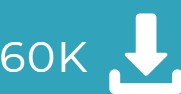

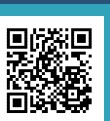

# POPULAR PROJECTS

## FinRL

FinRL provides a full-stack pipeline to automatically streamline the development of trading strategies and help overcome the steep learning curve. It has a three-layer software stack and facilitates a faster iteration of trading strategies.

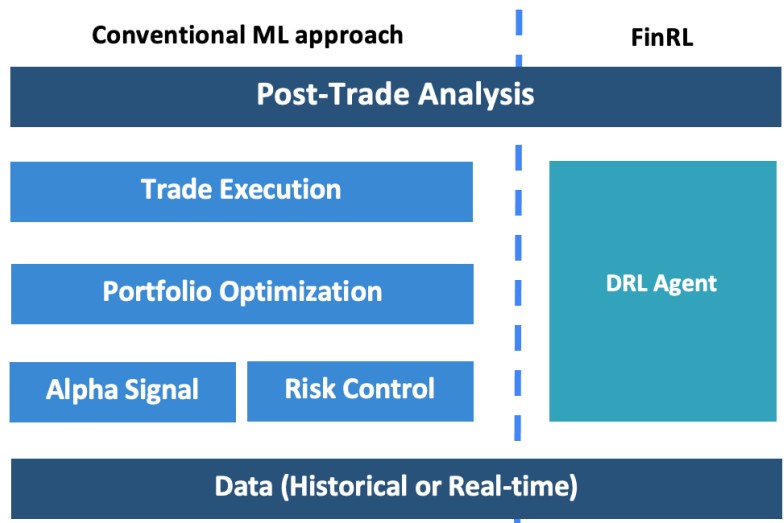

[1] FinRL: Deep reinforcement learning framework to automate trading in quantitative finance. ACM International Conference on AI in Finance (ICAIF), 2021.

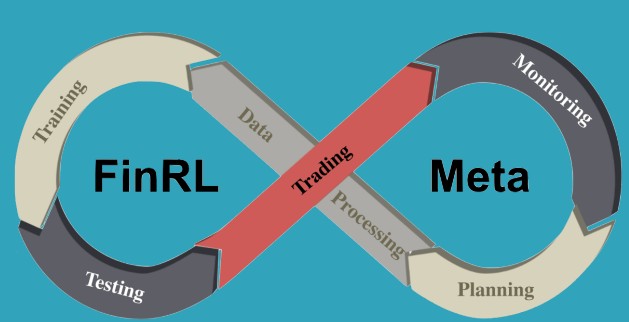

## FinRL-Meta

We provide a universe of market environments and benchmarks. It follows the DataOps paradigm and implements an automated pipeline: task planning, data processing, training-testing-trading, and monitoring the performance. It processes dynamic datasets from real-world markets.

[2] FinRL-Meta: Market environments and benchmarks for data-driven financial reinforcement learning. NeurIPS 2022, Datasets and Benchmarks Track.

## ElegantRL

ElegantRL is a cloud-native DRL library with high scalability and elasticity. We employ a tournament-based ensemble training: a leaderboard (of good agents) interacts with a training pool (of pods) in an asynchronous manner; an orchestrator coordinates the training process.

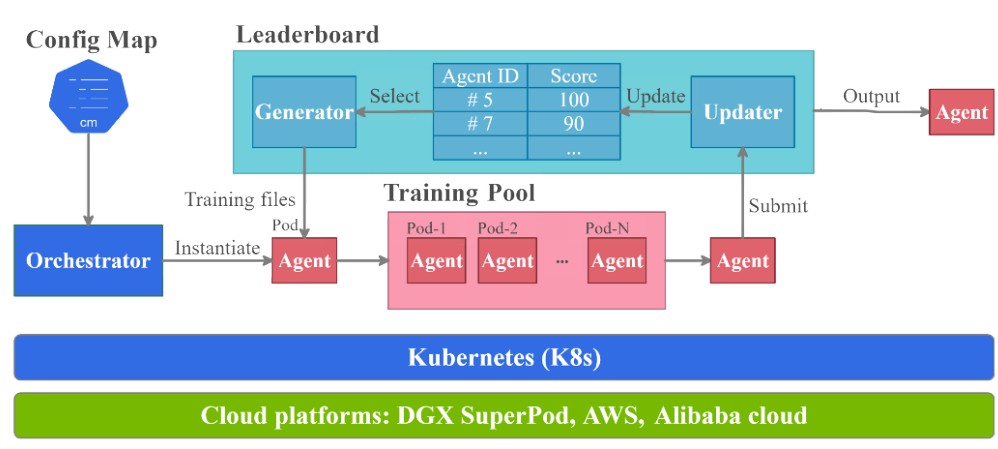

[3] ElegantRL-Podracer: Scalable and elastic library for cloud-native deep reinforcement learning. Deep Reinforcement Learning Workshop, NeurIPS 2021.

## "We build an open finance culture"