# OpenReview forum: "FinRL-Meta: Market Environments and Benchmarks for Data-Driven Financial Reinforcement Learning"
_NeurIPS.cc/2022/Track/Datasets_and_Benchmarks — NeurIPS 2022 Datasets and Benchmarks _

### Official Review · Reviewer_hKNR · 2022-07-03

**Rating:** 7
**Confidence:** 3

**Strengths:**

**Data diversity**. Connecting a wide variety of sources is a major contribution. This opens the door to easily benchmarking approaches on many tasks.

**Real world task**. Enabling near real-time paper trading (and related activities) provides an important step toward "real world" validation. This is relevant to the broader RL community as it provides a rare case of a task that can be tested against real world performance without major investment: robotics requires simulation or expensive equipment and games are only available in simulations against other computer agents (not human adversaries). Creating the infrastructure to support this task is a meaningful contribution toward getting RL in the real world in the hands of researchers.

**Interface generality**. Adopting OpenAI gym allows easy integration with many RL algorithms. Connectors facilitate this with widely used RL frameworks. This makes the benchmark readily accessible to RL researchers and relatively easy to learn for those without RL background.

**Documentation**. The repository includes extensive tutorials and documentation. Some seem to be out of date, but the large and active community can help mitigate against this.

**Weaknesses:**

**Domain terminology**. As a researcher outside the domain of finance much of the terminology was new and confusing at times. It would help to have some of the key terms introduced sooner or a glossary to help new researchers getting up to speed in the domain.

**Additional Feedback:**

I was unable to run the tutorials on a Windows 11 machine or using Colab to directly run the tutorial notebooks. I specifically tried the introductory "Stock_NeurIPS2018.ipynb" and "China_A_share_market_tushare.ipynb". Both colabs reported a missing notebook. Locally I was unable to install using the pip install from github (the first command). Is there an alternative or recommended way to try things?


# Questions
(Some are repeated from above to gather in one place)

**[Q0]** What is lacking or different from NeoRL? The text does not describe any specific differences:
> A stock trading environment from FinRL [26, 27] is included, however, it cannot satisfy the increasing demands from the rapidly growing community.

**[Q1]** Are GPUs currently supported? The text was ambiguous and the website mentions GPU support as a future goal.

**[Q2]** How long are the typical lags imposed when trying to process data in real time? Could this be added to Table 3 (in the supplement)? This would be helpful for users when working with environments that have high frequency updates.


# Feedback
Below are minor comments and suggestions to consider.

Figure 6: The Sharpe ratio line is hard to read. Consider plotting this on a separate plot or using a different color (perhaps black) to improve the contrast.

> "We replace the financial turbulence index with the volatility index (VIX) that can be accessed immediately."
Please provide some definition of these indexes or a reference. Coming from outside the domain I was unable to know how they differ.

**Clarity:**

Generally yes. It would help to define some terms earlier in the paper. Some from my notes: information leakage, backtesting stage, Sharpe ratio (at least the interpretation / meaning), alpha signals, mart beta index, ETF (the acronym alone is used).

**Correctness:**

The dataset processing is a core part of the paper contribution and appears to be correct. Specifically, the work addresses weakness of prior methods that do not facilitate testing against "live data".

Evaluations replicate prior papers and are done appropriately. These show baselines leave room to improve on the task. The experiment designs all reflect standard practices.

**Documentation:**

The code provides interfaces for users to query data, rather than a static set of prepared data. Maintenance and availability are thus subject to the upstream data sources themselves, with the code to interface with these being open source. Ethical uses are not directly addressed in detail, but the domain (financial trading) poses few risks.

The provided datasheet covers the main uses.

Maintenance ultimately hinges on the community. This is a large community backed by a non-profit organization, so is likely to persist.

**Ethics:**

No.

**Relation To Prior Work:**

Yes (mostly). Previous RL environments for finance are out of date and not being updated. None seem to have as wide an array of data sources to interact with.

One aspect that was not clear is the relation to NeoRL:
> A stock trading environment from FinRL [26, 27] is included, however, it cannot satisfy the increasing demands from the rapidly growing community.

What specifically is lacking?

**Summary And Contributions:**

The paper provides a set of python APIs to gather and process a wide variety of financial data sources that are then exposed to users through the widely adopted gym reinforcement learning interface. Interfaces allow users to plug into widely adopted existing RL agent training frameworks to employ baseline algorithms or develop new algorithms. Data sources cover a variety of trading markets spanning US stocks, Chinese securities, cryptocurrency exchanges, and foreign exchanges (with many options per category). The data processing layer is designed to allow near real-time interaction with these environments and include extensive options for data cleaning and feature engineering. Tutorials cover many different environment types and features of the benchmark. Experiments show the differing performance of RL agents against baseline trading strategies, demonstrating the potential for improvement on these tasks. The benchmark is supported through a research community that is maintaining and developing the work.

---

### Official Review · Reviewer_qzUQ · 2022-07-23
**Good interface to benchmark and develop RL methods on real-world financial market environments**

**Rating:** 7
**Confidence:** 3
**Clarity:** The paper is well written.

**Strengths:**

The main strength of the work is the relevance of the library to the financial RL community. The library fills an important void and has the potential to become (if it is not already) the de facto standard for financial RL research.

**Weaknesses:**

While the work and the contribution of the library are certainly significant, I personally see two main weaknesses that prevent me from giving a higher rating. I will be willing to increase my rating after discussion with the authors.

First, to the best of my understanding the library does not provide environments with new simulated scenarios, but only historical data. I wonder whether this is enough to close/reduce the simulation-to-reality gap. Indeed, the RL agent will see a limited number of possible scenarios, with the omnipresent risk of overfitting. When deployed to real-world, the RL agent might not be prepared for the possible new scenarios. Are environments with new simulated scenarios considered for future integration in the library? I apologize if this is already present and I missed it.

Second and partially connected to the first point is the full observability assumption. In the paper, the environments are described as MDPs, namely the state that the agent perceives contains full information of the environment state. However, there is abundant literature that considers financial markets as hidden Markov models, namely the observations contain noise-corrupted information of the true market hidden states. This is also acknowledged by the authors that highlight several times the low signal-to-noise ratio of financial data. I would have expected the authors to model the problem as POMDPs.

**Additional Feedback:**

The sentence in Section 3.1:
"Training and testing environments based on historical data may not simulate real markets accurately due to the simulation-to-reality gap [12, 13], and thus a trained agent cannot be directly deployed in real-world markets."
sounded a bit strange to me since the purpose of the library is to provide high-quality benchmarking environments, which aim to reduce the simulation-to-reality gap. Are not FinRL-Meta training and testing environments based on historical data?

Despite the aforementioned points that require some discussions, I think that the overall work is great and the library will be very helpful for the community.



**Correctness:**

The overall work seems correct and constructed in a solid way, except perhaps for the aforementioned MDP - favored over the POMDP - assumption.

**Documentation:**

The repository is very active, it seems well maintained and it has extensive documentation with the presence of tutorials.

**Relation To Prior Work:**

The authors clearly discussed how this work differs from previous contributions.

**Summary And Contributions:**

The works proposes FinRL-Meta, a library to ease research and applications of RL methods in financial market environments. The authors develop an interface, that substantially facilitates access, cleaning, and preparation of historical structured and unstructured financial data, which constitutes the state that the RL agent perceives. Agent-environment interactions are standardized to common libraries that provide RL environments and a multitude of classical financial tasks are possible (stock trading, portfolio allocation, etc..). The library provides large flexibility to model the investor behavior (e.g., risk aversion or whether to allow short-selling) and real-world markets (e.g. transaction costs). The combination of task, input data, and possible actions gives rise to a vast array of possible environments.

The main contribution is certainly to provide the RL community with a structured interface to ease research in financial reinforcement learning. The authors also provide benchmarks and tutorials of common RL methods as baselines and for educational purposes.

---

### Official Review · Reviewer_Cw3U · 2022-07-24
**more like software manual than a paper**

**Rating:** 6
**Confidence:** 3
**Correctness:** yes

**Strengths:**

- the authors introduce the concept of DataOps into this area
- the authors incorporate some papers as benchmark to verify the environment
- the submission could be connected with a wide range of data sources
- the submission considers practical factors of markets, such as the factors listed in Sec. 4.2
- the submission takes the pipeline (workflow) into account (Fig.2)

**Weaknesses:**

The authors target a promising area and aims at the important and right problem, which is to make the RL algorithms in FinTech (esp. Quant) more practical and workable in the real-world scenarios. The authors provide a lot of function descriptions and features, but it is not enough to give me a clear answer that how this submission could achieve this goal. I am open to update my rating if the author could address my concerns.

- How the environment works with the accessed market data. What's the internal price-volume generation mechanism after an action is taken? It will continue the historical data, fitted to an empirical response formula, or generated by the Level-2 market data? How are the states and observations generated from the accessed data? Is there an extension interface (e.g., callbacks or hooks) for users to control such processing?

- It's fantastic to leverage finance big data. However, there is no detail other than mentioning the category of data in Fig. 3. Take the news sentiment data as an example, is it just an indicator connected with a data service provider, or an interface that user needs to prepare news and DNN models individually? How to handle the time alignment and avoid information leakage from news data?

Above are just two examples. In fact, there are huge research and engineering challenges in the financial RL area. I believe this submission has done a lot of solid work to mitigate the difficulties, but they're not well demonstrated in the paper (esp. Sec. 4).


**Additional Feedback:**

- About the status of the functions listed in Fig. 3. Have all these functions or features available or are some of them just in the roadmap?

**Clarity:**

- This submission is more like a software manual than a paper to me

**Documentation:**

yes

**Relation To Prior Work:**

the comparison with prior works is not very clearly demonstrated

**Summary And Contributions:**

1. The authors introduce the DataOps practice and provide market environments through an automatic pipeline that collects dynamic datasets from real-world markets and processes them into standard gym-style market environments
2. The authors incorporate some papers as benchmarks and provide tutorials and demos in this area

---

### Official Review · Reviewer_tYqY · 2022-07-28
**Great financial reinforcement learning framework**

**Rating:** 9
**Confidence:** 1
**Correctness:** Correct
**Clarity:** Yes

**Strengths:**

1. It provides an easy-to-use financial reinforcement learning framework.
2. It includes sufficient market data and side information for users to play with.
3. Commonly used baselines are also provided in the training framework.


**Weaknesses:**

The tutorials are not fully developed (https://finrl.readthedocs.io/en/latest/tutorial/2-Advance.html)

**Additional Feedback:**

N/A

**Documentation:**

Yes

**Ethics:**

There is no ethical concerns

**Relation To Prior Work:**

It is clearly discussed

**Summary And Contributions:**

This paper provides a high-quality, easy-to-use, and complete framework for financial reinforcement learning. It considers sufficient information resources and multiple baselines. Overall this paper is interesting and makes a big contribution to the research community.

---

### Official Review · Reviewer_DPoU · 2022-07-28
**Comprehensive library for reinforcement learning in finance for users of all levels**

**Rating:** 9
**Confidence:** 4

**Strengths:**

The authors provide an extremely robust library and approach to guiding us to it in this paper.  They follow strong software engineering and data science practices to build FinRL-Meta and have also provided lots of data from varied sources.  There is a structured curriculum for users to learn and to follow each example.  Assumptions are clearly state in each reproduction of a task in their environment, with nicely selected examples and clear descriptions of the application.  The paper is also extremely well organized but covers all of the special considerations the authors put into their work without being arduous to read.  The chosen reproductions and trading performance shows a strong understanding of metrics that are often seen in financial market prediction problems.

**Weaknesses:**

The paper reads as an extended guide to the library, the authors' main contribution.  The limitations I point out are specific to certain areas, where I think more detail can be given even if it's not part of the authors' contribution, as it is important to demonstrate the environment/background where users will operate in.  That being said, specific contributions can be highlighted more clearly (vs reproduced experiments if not designed by the authors) particularly in section 5.

It also would be nice to have a simple homegrown example figure (probably already exists in some of the tutorial notebooks) but even in the appendix, highlighting a base usage with a common data set would be very valuable (think iris + any regression in sklearn).  Maybe this can be in the form of extending a little bit more discussion on the results/rationale of different methods in "Stock trading task"

Specifically, some examples I'd like additional detail on:

Intro: "Simulation to reality gap" is a problem not just for deep reinforcement learning but all ML, what are some steps the field has taken to address this?

5.1 Should include Sortino ratio as a common metric which modifies Sharpe to focus on the downside variance

Figure 5 reproductions - could talk a little bit more in the caption or text about what the graphs are trying to achieve (will also be seen as examples of what can be done) currently it's taken on word that the graphs look "reproduced"

Section 5 Explain some possible strengths and weaknesses of the separate learners that get ensembled together?

Is there any code in the library that helps provide visualization?



**Additional Feedback:**

Some additional grammar/fluidity comments:

Abstract: Second sentence remove "however" as it supports the claim of the first sentence
Abstract/Intro: "is challenging"
3.1 Second to last bullet - what is a "busted" stock?
3.1 Last Bullet - existing "research"
Figure 4 - should the second layer of tutorial read "Advanced" instead of "Advance"
Appendix B1 1st Bullet "as data is refreshing"
Appendix B5 5th Bullet "problems"
Appendix B6 1st Bullet "will always be held"

General: "code" is understood to be plural
General: some usage of acronyms like OHLCV, might not be known by non-finance readers

**Clarity:**

The paper is extremely well structured and reads fluently.  The supplementary appendix written in the form of an FAQ works well to answer common questions that would come up in terms of dataset, usage and cautions.  As mentioned before, this paper reads differently due to its contribution in the form of a product - a library to be used by others.

**Correctness:**

Given the contributions are in the form of a library and framework, correctness quantified would be whether the library works.  This is demonstrated by reproducing the examples of literature.  The broader discussions about both the state of reinforcement learning in finance as well as broader financial topics/common approaches gives me confidence that the authors are knowledgeable in the space and thus I do not doubt correctness.

**Documentation:**

Documentation is extremely clear and provided on github.  Part of the contribution is the tutorials on top of the created library/environment, so this is done well by construction.

The appendices in the form of an FAQ helps users utilize their library, and they emphasized the importance of reproducibility (and highlighted it as a current weakness of the field).

The data sets themselves are also thoroughly documented as part of the appendix.

**Ethics:**

No concerns

**Relation To Prior Work:**

Prior work is addressed in section 2 and as part of the introductory section.  The FinRL library they are building on top of us documented and maintained by publicly.  In particular, the authors create a very structured library that is well documented with features, environments to enable financial learning.  They also provide documentation and learning tutorials for users.
 This is their value-add contribution.  The past methods of reinforcement learning strategies were reproduced so this is another flavor of relation to other works.  Lastly, all libraries and datasets are built off the work of others, all of which are comprehensively cited and referenced.

**Summary And Contributions:**

The authors have created FinRL-Meta, an environment built on top of FinRL that scaffolds more financial market specifics for the user.  Their contributions include providing a structure organized via DataOps paradigm with pre-built environments and data preprocessing, a reproduction of other paper's models using this aforementioned framework, and finally an organized educational curriculum for users to learn how to use the environment.  All 3 pieces integrate into one polished product, with a couple future steps also laid out.

---

### Official Review · Reviewer_uJbg · 2022-07-29

**Rating:** 7
**Confidence:** 4
**Correctness:** Yes, the reproduction of prior papers…
**Clarity:** The paper is clear.

**Strengths:**

- The library is well-built, clear, and maintained. It's a convenient tool for both the research community and everyday use.

**Weaknesses:**



**Additional Feedback:**

N/A

**Documentation:**

The GitHub repo is well-documented, and the Jupyter notebooks are useful tutorials.

**Relation To Prior Work:**

The authors mention the market environments and their shortcomings.

Why does NeoRL not satisfy the demands from the rapidly growing community?

**Summary And Contributions:**

- FinRL-Meta Library, which provides hundreds of market environments with dynamic datasets from real-world markets.
- A collection of performance of different models on the market environments.
- Jupyter notebooks as tutorials.

---

### Meta-Review · Area_Chair_Zni3 · 2022-09-06

**Recommendation:** Accept
**Confidence:** 4

**Metareview:**

The authors propose a major software framework, FinRL-Meta, for testing reinforcement learning (RL) algorithms on financial data.

At the core the benchmark contains an extensible set of data sources and market simulation environments and is connected to three state-of-the-art Deep RL libraries.
Furthermore, the authors demonstrate the viability of their system through reimplementations of import papers in the space.

The benchmark also contains a large number of tutorials and how-to's to allow newcomers get up to speed quickly and seems to be overall extremely well documented.

My only concern is that there might be potentially too much flexibility in the benchmark, leading to a possible fragmentation of research questions.

---

### Decision · Program_Chairs · 2022-09-16

Accept